# Indigenous Australian women's experiences of participation in cervical screening

**Tamara L. Butler**[1]*, **Kate Anderson**[1], **John R. Condon**[1], **Gail Garvey**[1], **Julia M. L. Brotherton**[2], **Joan Cunningham**[1], **Allison Tong**[3], **Suzanne P. Moore**[1], **Clare M. Maher**[4], **Jacqueline K. Mein**[5], **Eloise F. Warren**[6], **Lisa J. Whop**[1]

1 Wellbeing and Preventable Chronic Diseases Division, Menzies School of Health Research, Charles Darwin University, Darwin, Northern Territory, Australia, 2 VCS Population Health, VCS Foundation, Melbourne, Victoria, Australia, 3 Sydney School of Public Health, The University of Sydney, Sydney, New South Wales, Australia, 4 Southern Queensland Centre of Excellence in Aboriginal and Torres Strait Islander Primary Health Care, Brisbane, Queensland, Australia, 5 Wuchopperen Health Service Pty Ltd, Manoora, Queensland, Australia, 6 Yerin Eleanor Duncan Aboriginal Health Centre, Wyong, New South Wales, Australia

* tamara.butler@menzies.edu.au

## Abstract

Aboriginal and Torres Strait Islander (collectively, Indigenous Australian) women experience a higher burden of cervical cancer than other women. The National Cervical Screening Program (NCSP) is failing to meet the needs of Indigenous Australian women, resulting in many women not regularly participating in cervical screening. However, one third of Indigenous Australian women do participate in cervical screening. The reasons that some women in this population commence and continue to screen remain unheard but could provide insights to support women who currently do not participate. We aimed to describe Indigenous Australian women's experiences and views of participation in cervical screening by yarning (a culturally-appropriate interview technique) with 50 Indigenous Australian women aged 25–70 years who had completed cervical screening in the past five years, recruited via Primary Health Care Centres (PHCCs) from three jurisdictions. Aboriginal or Torres Strait Islander women researchers conducted the interviews. Thematic analysis identified six themes: screening as a means of staying strong and in control; overcoming fears, shame, and negative experiences of screening; needing to talk openly about screening; the value of trusting relationships with screening providers; logistical barriers; and overcoming privacy concerns for women employed at PHCCs. Despite describing screening as shameful, invasive, and uncomfortable, women perceived it as a way of staying healthy and exerting control over their health. This ultimately supported their participation and a sense of empowerment. Women valued open discussion about screening and strong relationships with health providers. We identified logistical barriers and specific barriers faced by women employed at PHCCs. This study is strengthened by a research approach that centred Indigenous Australian women's voices. Understanding the experiences of Indigenous Australian women who participate in screening will help screening providers support women to start and continue to screen regularly. Recommendations for practice are provided.

**Data Availability Statement:** Data cannot be made publicly available for ethical reasons. Participants did not consent to public release of full transcripts and the nature of the qualitative data are such that

public availability would compromise participant confidentiality. Requests for specific excerpts relating to published verbatim quotes will be considered on a case-by-case basis. Requests for data may be sent to the corresponding author (tamara.butler@menzies.edu.au) and to the Human Research Ethics Committee (HREC) of the Northern Territory Department of Health and Menzies School of Health Research (ethics@menzies.edu.au; reference number: 2017-2993), the institutional HREC committee. Please note that the authors will need to seek approval from the five HRECs listed in the Ethics Approval section before releasing data.

**Funding:** This study was supported by the National Health and Medical Research Council (NHMRC) funded Centre of Research Excellence (CRE) in Targeted Approaches To Improve Cancer Services for Aboriginal and Torres Strait Islander Australians (TACTICS; #1153027), the NHMRC-funded CRE in Discovering Indigenous Strategies to improve Cancer Outcomes Via Engagement, Research Translation and Training (DISCOVER-TT; #1041111), and the Cancer Council NSW Strategic Research Partnership to improve cancer control for Indigenous Australians (STREP Ca-CIndA; SRP 13-01, with supplementary funding from Cancer Council WA). The funders had no role in study design, data collection and analysis, decision to publish, or preparation of the manuscript. The views expressed in this publication are those of the authors and do not necessarily reflect the views of the funder. TB was supported by an Australian Research Council Discovery Australian Aboriginal and Torres Strait Islander Award (#IN190100050) funded by the Australian Government. KA, JCo, JB, SM, JM, EW, and CM received no specific funding for this work. GG and LJW were funded by NHMRC Early Career Fellowships (#1105399 and #1142035, respectively). JCu was funded by an NHMRC Research Fellowship (#1058244). AT was supported by a NHMRC Career Development Fellowship (#1106716).

**Competing interests:** The authors have read the journal's policy and have the following conflicts: The study is affiliated with Southern Queensland Centre of Excellence in Aboriginal and Torres Strait Islander Primary Health Care, Wuchopperen Health Service Pty Ltd and Yerin Eleanor Duncan Aboriginal Health Centre. These commercial affiliations provided support in the form of salaries for authors CM, JM, and EW, respectively. This does not alter our adherence to all the PLOS ONE policies on sharing data and materials.

# Introduction

## Cervical screening and cervical cancer in Australia

Cervical cancer is preventable through regular cervical screening tests and human papillomavirus (HPV) vaccination. Australia's National Cervical Screening Program (NCSP) has been in place since 1991. As a result of the nationally coordinated approach to cervical screening cervical cancer incidence in Australia has more than halved, with the age-standardised incidence rate declining from 18 new cases in 1990 (before the NCSP) to about 7 new cases for every 100, 000 women in 2002 and this has remained steady to 2015 (the most recent year for which data is available) [1].

In 2017, the program transitioned from two-yearly Papanicolaou (Pap) smears for women aged 20–69 years to five-yearly screening using HPV nucleic acid testing with partial genotyping and reflex cytology for women aged 25–74 years [1]. Adolescents aged 12–13 years receive the HPV vaccination via a school-based immunisation program, introduced for girls in 2007 (with a female catch-up program to age 26 years) and extended to boys in 2013 [1]. With high uptake of population-based HPV vaccination and HPV-based screening, modelling studies suggest that Australia may be among the first countries to reduce cervical cancer incidence to the "rare cancer threshold" of 6 new cases per 100,000 women as soon as 2020 and to meet the World Health Organisation's proposed elimination threshold of cervical cancer as a public health problem (<4 per 100,000) by 2028 [2, 3].

Despite Australia's success, there remain significant and persistent disparities in rates of cervical screening and cervical cancer outcomes for Aboriginal and Torres Strait Islander women (hereafter, respectfully referred to collectively as Indigenous women). This indicates that the NCSP does not meet the needs of Indigenous women, and that these disparities in access must be addressed to ensure that the achievement of elimination targets includes Indigenous Australian women [4]. Indigenous Australians account for 3.3% of Australia's total population [5]. The age-standardised incidence of cervical cancer for Indigenous women aged 20–69 in five Australian states or territories in 2011–2015 was 22.3 per 100,000 women [1], comparable to rates in low and middle income countries [6]. Compared to non-Indigenous women, Indigenous women are 2.5 times more likely to be diagnosed with cervical cancer and are 3.8 times more likely to die of cervical cancer [7]. Indigenous women participate in the NCSP at a lower rate than non-Indigenous women [1]. In Queensland, the estimated 2010–2011 age-standardised two-year participation rate was 33.5% for Indigenous women compared to 55.7% for non-Indigenous women [8]. Similar estimates of Indigenous and non-Indigenous women's participation have been found for women living in rural and remote Queensland communities [9] and in the Northern Territory [10]. To address these disparities, we need to understand both the individual and structural barriers and enablers to participating in the cervical screening program.

## Barriers and enablers to cervical screening

A synthesis of qualitative evidence reporting predominately non-Indigenous women's experiences of cervical screening in countries with an organised program (including Australia) found several barriers to participation, including: myths about the causes of cervical cancer; beliefs about cervical cancer as an older woman's illness; a lack of symptoms and feelings of health; absence of family history; and perceptions of screening as unimportant [11]. Screening was viewed as an emotionally and physically taxing event due to fear of receiving abnormal results or a cancer diagnosis and perceptions of sexual promiscuity. Women perceived the procedure as intrusive, prompting feelings of embarrassment, shame, and vulnerability, exacerbated in the presence of male health professionals [11]. A systematic review focusing on barriers to screening for Australian women reported similar factors relating to personal barriers (e.g., lack of

knowledge), practitioner barriers (e.g., gender of practitioner), the screening test (e.g., pain, discomfort, and embarrassment), and logistical barriers (e.g., lack of time) [12].

Importantly, screening barriers and enablers have been linked to past screening behaviour: Australian women overdue for screening were more likely to report embarrassment as a barrier to screening, while screened women were more likely to report having a regular general practitioner and the low cost of the test as facilitators of screening [13].

Indigenous women in Australia face additional structural barriers to cervical screening, which demonstrate that the Australian cervical screening program does not meet the needs and preferences of Indigenous women. Where structural barriers exist, systemic disadvantages persist. While not experienced by all Indigenous women, barriers may include: racism; a lack of cultural awareness and sensitivity among health professionals; a lack of culturally appropriate communication available through reminder messages and promotional material; ongoing impacts of colonisation and imposed institutional control and mistreatment leading to a distrust of health services and institutions; perceived lack of confidentiality in services because of kinship relationships; health professionals' use of medical jargon; gender of the health professional; and practical issues including distance to the health service, transport, and financial burden [14–16]. Many of these barriers are shared with other Indigenous peoples worldwide [17–20]. Furthermore, some Indigenous Australians' fatalistic views, beliefs that cancer is contagious and a form of retribution for wrongdoings, feelings of shame, and misunderstandings about the purpose of cancer screening influence their engagement with health services and in cancer prevention programs [21, 22]. These views and beliefs stem from failings of the Australian health care system to engage appropriately with Indigenous Australians.

Factors that enhance Indigenous women's access to cervical screening are documented less, but evidence suggests that some enabling factors include: having access to a dedicated women's health program and a choice of seeing a female practitioner (especially in remote areas); accessibility of health services; having trust in the health practitioner; fear of developing cancer after the death of kin from cervical cancer; and having had a past positive cervical screening experience [15].

## The current research

Consistent with the factors enhancing access to cervical screening, evidence from Queensland indicates that approximately one-third of Indigenous women regularly participate in screening [8]. Whop et.al. reported most Indigenous women who had participated in screening in 2010–2011 had completed screening at least once in the 10 years prior, suggesting that once women start screening, they are likely to continue to screen [8]. While research has justifiably focused on the barriers to women's participation, it is important to understand the perspectives of Indigenous women who participate in screening to better understand the factors involved in women commencing and continuing to screen. Such an approach allows researchers, clinicians, and practitioners to identify the individual and structural factors supporting women's participation that should be maintained, enhanced and/or implemented. Understanding the individual and structural factors that support Indigenous women's participation may also shed light on factors that do not support Indigenous women's participation. As such, the current research aims to describe the views and experiences of Aboriginal and Torres Strait Islander women who participate in cervical screening.

## Research methods

Reporting of the research adheres to the Consolidated Criteria for Reporting Qualitative Research (COREQ) [23].

## The *Screening Matters* study approach

Data reported here are a subset from a larger study ("*Screening Matters*: *Aboriginal and Torres Strait Islander women's attitudes and perspectives on participation in cervical screening*") in which women were invited to yarn about cervical screening. For *Screening Matters*, we recruited two groups of women based on their cervical screening behaviour in the last five years: those who had screened and those who were under-screened or never-screened. Women were eligible to participate in *Screening Matters* if they identified as Aboriginal and/or Torres Strait Islander and were between 25–74 years old. Women were excluded if they had had a hysterectomy. We aimed to yarn with a minimum of 10 women (5 screened women and 5 under-screened or never-screened women) at each participating Primary Health Care Centre (PHCC). Health care providers also participated in interviews about their views on cervical screening delivery for Indigenous women. We report the views of screened women only in this paper.

The *Screening Matters* study was conceptualised, led, and conducted by Indigenous Australian women: LJW, TB, GG, BM and two Aboriginal community research officers. It privileged the voices of Indigenous Australian women–the participants. Finally, the study aimed to understand the individual, community, and structural influences on Indigenous Australian women's participation in cervical screening. Together, this approach ensured that Indigenous Australian women's perspectives on cervical screening were centred in the research.

## Participants and recruitment

Data reported in this paper were from the 50 women in the *Screening Matters* study who self-reported that they had completed cervical screening within the last five years. Women were recruited using a convenience sampling approach by staff working at five PHCCs across Queensland, New South Wales and Northern Territory. The PHCCs included in this study are Aboriginal Community Controlled Health Services or Government-run clinics that serve a large proportion of Indigenous clients. Participant recruitment methods were determined by each PHCC, including staff speaking to women about the study or via women's groups' social media pages. The number of screened women recruited ranged from 6 to 14 per PHCC.

## Data collection

Women received information about the study from both PHCC staff and the project staff. After providing written informed consent to participate in the study, women completed a short demographic and health survey. We used an Indigenous qualitative methodology called yarning [24–26]–a culturally-appropriate process of sharing knowledge. Yarning involves the sharing, listening, interpreting, re-interpreting and making sense of past events to ensure cultural survival in the present day [24–26].

All yarns were conducted by an Aboriginal or Torres Strait Islander woman (LJW, TB, BM, and two anonymous community research officers). In one PHCC, the *Screening Matters* project team were advised that it would be more appropriate and comfortable for participants if staff members known to women conducted the yarns. Two Aboriginal research officers on the PHCC staff were trained as researchers by the project team. In the remaining four PHCCs, *Screening Matters* project staff conducted yarns to maintain consistency of approach and reduce workload burden on PHCC staff.

The yarns followed a question guide, which provided topics and questions that should be covered but that allowed participants freedom in how they told their stories. At the end of the interview, questions were included about the option for self-collection and perceptions of the self-collection instruction guide, both part of the new cervical screening program

recommendations in Australia. As the current analysis focuses on women's perceptions of clinician-collected cervical screening, these data are not included in the analysis reported here. S1 Text presents a modified list of questions relevant to the current analysis.

Most participants had no pre-existing relationship with the person conducting the yarns, except for women at one PHCC who yarned with the two community research officers. Participants knew the researcher officers because they visited the community regularly as community education outreach officers. Yarns were conducted with women from April to November 2018 in a private space in the PHCC. Most yarns were conducted between the researcher and the participant, but in some cases, women were comfortable with the presence of other women during the conversation, either as another participant in the study, support person, or observer. Yarns took between 10 and 45 minutes to complete. Researchers took field notes and regularly discussed with each other privately how the yarns were progressing and the emerging themes. Yarns were audio-recorded with the participants' consent. If consent to record was not provided, the interviewer wrote notes during the yarn and added detail after it had concluded. Data on women who declined to participate in the study was not collected.

Data saturation (no new themes raised by participants) was assessed at the point of data analysis, rather than during data collection. This was due to recruitment being contingent on the number of participants PHCC staff recruited, budgetary limits, and our focus on recruiting a diverse range of participants. During data analysis, it was clear that there were no new, unexplored themes raised by participants.

## Data analysis

Participants' residential geographic remoteness were categorised by converting postcodes to Remoteness Areas using the 2016 Australian Statistical Geography Standard [27]. Three levels were used for reporting: major city, regional, and remote. Yarns were transcribed verbatim and imported into NVivo (QSR International Pty Ltd, version 11 [28]) for thematic analysis. Transcripts were not returned to participants for comment. Thematic analysis was iteratively conducted through line-by-line review of the transcripts to inductively identify themes. Two authors (TB and KA) analysed the data in three stages. First a random selection of 12 yarns were read to develop an initial coding structure, which was discussed with *Screening Matters* investigators. We then iteratively adjusted the coding structure based on this feedback until consensus was reached. The remaining 38 yarns were split between the two authors for further analysis, with regular meetings to discuss emerging themes and amend themes as necessary. Finally, the first 12 yarns were re-analysed to ensure consistency with the final coding structure. Indigenous researchers conducted the analysis (TB) and provided input on interpretation of findings (LW, GG) with support from an experienced non-Indigenous qualitative researcher (KA). Themes were then organised into larger thematic findings, which are reported below. These themes were not validated by the participants.

## Ethics approval

Ethics approval for this research was obtained from the Aboriginal Health and Medical Research Council of New South Wales (AH&MRC) Ethics Committee (1341/17), Central Australian Human Research Ethics Committee (CAHREC, CA-18-3113), Far North Queensland Human Research Ethics Committee (FNQ HREC, HREC/18/QCH/41-1218), Human Research Ethics Committee of the Northern Territory Department of Health and Menzies School of Health Research (2017–2993), and Metro South Human Research Ethics Committee (MSHREC, HREC/18/QPAH/52). Informed written consent was obtained from all participants.

## Results

In total, the *Screening Matters* study recruited 79 eligible participants, of whom 50 were screened women (reported in this analysis). Characteristics of the screened participants are provided in Table 1. Most (88%) women identified as Aboriginal and the most common age group was 25 to 39 years (32%). Fourteen women were both clients and employees of the clinic.

We identified six themes through analysis of the yarns: screening as a means of staying strong and in control; overcoming fears, shame, and negative experiences of screening; needing to talk more openly about screening; relationship with screening provider is critical; logistical barriers to screening; and issues for women employed at PHCCs. Verbatim quotes are marked with the participant code (entire *Screening Matters* sample).

### Screening as a means of staying strong and in control

Across all geographic regions, participants reported a desire to live a strong and healthy life–often framed in terms of staying well and keeping strong, as opposed to avoiding illness. Participation in cervical screening made women feel empowered and that they were taking control over an important aspect of their health. This was in contrast to other aspects of women's health feeling somewhat uncontrollable, such as having periods and giving birth. Women reported that participation in cervical screening gave them and their family peace of mind in the knowledge that they were healthy and cancer-free. Women frequently recognised the benefits of screening for the early detection and prevention of cancer, keeping healthy, and an opportunity to check for general health issues relevant to the area including sexually transmitted infections.

Women wanted to stay healthy and live long lives not just for themselves, but with reference to being present for children, grandchildren, and family, and outliving previous generations of family who died young due to cancer. Women wanted to be healthy so they could fulfil caring duties and responsibilities as part of valued roles in family and community.

> *It's about listening to my body. It's about trying to keep it healthy for my children. I want to live as long as I can to bring them up to see them. Not only that, I want to feel good too. And I have had friends who have been diagnosed and that motivates me also to do it. But all in all I do it for myself and my family.* P03

While women recognised the potential for screening to save lives, some felt resigned to the fact that cervical screening was an unpleasant but necessary part of life. Some women reported a 'just do it' attitude to screening, referring to the short duration of the test as a positive factor. On the other hand, others viewed screening as routine to the maintenance of a healthy body.

> *It's just what you do. I brush my teeth. It keeps coming down to that because it really for me is just about general body maintenance, you brush your teeth, you eat your food, I take my tablets. . . I do what I have to do.* P35

Women wanted to pass on knowledge about cervical screening to other women in their lives as a means to enhance their control over health. Women expected that young girls learned about screening and other women's health issues from their mothers as part of growing up strong. Women reflected on the information they received about women's health matters from their mothers and were motivated to improve the intergenerational transmission of information going forward.

**Table 1. Participant characteristics.**

|  | Number of participants | Per cent of participants |
|---|---|---|
| *Indigenous identification* |  |  |
| Aboriginal | 44 | 88 |
| Torres Strait Islander or both Aboriginal and Torres Strait Islander | 6 | 12 |
| *Age group* |  |  |
| 25–39 years | 16 | 32 |
| 40–49 years | 13 | 26 |
| 50–59 years | 11 | 22 |
| 60+ years | 10 | 20 |
| *Had children* |  |  |
| Yes | 40 | 80 |
| No | 10 | 20 |
| *Presence of one or more chronic disease* |  |  |
| Yes | 29 | 58 |
| No | 21 | 42 |
| *Employed at Primary Health Care Centre* |  |  |
| Yes | 14 | 28 |
| No | 36 | 72 |
| *Education level* |  |  |
| Year 12 or below | 29 | 58 |
| TAFE certificate/diploma, trade certificate | 13 | 26 |
| University | 8 | 16 |
| *Marital status* |  |  |
| Single | 19 | 38 |
| Defacto/married | 24 | 48 |
| Separated or divorced | 7 | 14 |
| *Main language spoken at home* |  |  |
| English | 39 | 78 |
| English and another Aboriginal language | 9 | 18 |
| Missing | 2 | 4 |
| *Residential postcode remoteness* |  |  |
| Major city | 33 | 66 |
| Inner or Outer Regional | 10 | 20 |
| Remote or Very Remote | 6 | 12 |
| Missing | 1 | 2 |
| *State/territory* |  |  |
| New South Wales | 27 | 54 |
| Northern Territory | 6 | 12 |
| Queensland | 17 | 34 |
| *Had HPV Vaccination* [a] |  |  |
| Yes | 9 | 18 |
| No | 29 | 58 |
| Don't know | 12 | 24 |
| *Number of participants per PHCC* |  |  |
| PHCC 1 | 6 | 12 |
| PHCC 2 | 9 | 18 |
| PHCC 3 | 8 | 16 |

(*Continued*)

**Table 1.** (Continued)

| | Number of participants | Per cent of participants |
|---|---|---|
| PHCC 4 | 13 | 26 |
| PHCC 5 | 14 | 28 |

[a] Many women expressed that they were not confident about their vaccination status

> . . . I didn't get it explained too much about that growing up into a woman off my mum. Whereas I want to do the opposite with my daughter and sort of be one step ahead of her, prepared and ready. . .. having that information would be good to have it there for her if she ever does want to know about it. P45

To enhance other women's control over their health, women believed that conversations about screening should move beyond simple encouragement to participate, to helping women gain an understanding of *why* screening is beneficial.

## Overcoming fears, shame, and negative experiences of screening

Women described screening as: "*painful*", "*degrading*", "*invasive*", "*uncomfortable*", "*embarrassing*", "*daunting*" and "*scary*", and felt reluctant to participate in screening. They also felt anxious waiting for the screening results. Whilst some felt a sense of shame associated with participating in screening and talking about it, women overcame these emotional obstacles to complete screening, stating that the health benefits and sense of control screening afforded them outweighed the shame and discomfort surrounding the process:

> . . . it's no shame, you have to go and do your tests and all that. . . Every women [sic] in Australia have to do it every time, black or white. . .I don't get shame, because I want to look after myself and for my health too. P76

In some cases, women actively encouraged other women to overcome their fears, to "*not be shame*", and to look after their health by participating in screening. These women were motivated to set a good example by ensuring their own screening was up to date.

Women reported that shame about cervical screening was drastically reduced or eliminated after they had had children, leaving them feeling more relaxed about participating in screening.

> . . .Whatever the doctor says I just put my hand up, because after having children you don't care really. Just do what you have to do. P22

For many women having children was a critical turning point in their lifelong screening behaviour. Having children broadened many women's priorities to focus on staying healthy for their children and families. Capitalising on this, a participant employed as a health worker in the clinic reported that their maternal health team used post-birth check-ups to begin a conversation with new mothers to start screening. Another woman described a vivid experience which continued to shape her commitment to ongoing screening, where a health professional encouraged her to start screening by framing it as a central part of her new role as mother and primary carer for her child.

> I'll always remember that lady, and she was so nice too. And she's going, your job now is mummy, and you've got to be here, have this test every two years. And because that was at my

*six week check-up and I didn't want to have the pap smear, and she's going, you don't want to leave him. And I'm thinking, oh my God, because I was a single mother, oh my God, I've got to look after him. So I did; never missed it.* P35

Women's negative early experiences of Pap smears loomed large in their recollections and strongly influenced their decisions to continue screening and attitudes toward screening; as one woman said: "*the first one's always the worst. . .*" P56. Another woman described her negative first experience of screening after having a child, which led to her not participating in screening for a very long time:

*. . . I think it was [hospital name], and they were the roughest they could be in there. They weren't gentle at all, especially having something like that done and they say, "Lay down," and wham, you know, they're in, and that, and it was awful. I said, no, that's it.* P64

This woman recommended screening when she started attending her current PHCC and built a strong relationship with a doctor there.

One woman with a physical disability described feeling "*daunted*", "*overwhelmed*" and crying during her first Pap smear, due to the procedure not being explained to her beforehand. She reflected that because she looked old enough to have had a Pap smear before, "*I guess they thought that I would know what was going to come but I didn't.*" P71. Despite this experience, this woman explained that through a combination of family experiences with cancer and valuing looking after her health, she now views screening as necessary and important for women's health.

## Needing to talk more openly about screening

Women suggested that there was a need for women to talk amongst themselves about screening more openly to raise awareness and increase knowledge, saying that "*you don't hear about enough*" (P03). Some women expressed views that groups of women, such as younger women, older women, and women who had not had children, felt a strong sense of "*that shame factor*" (P04) about screening which made screening difficult to discuss in community. Open sharing of experiences, knowledge and information with other women was important in encouraging participation through reducing shame. It was important that this open discussion was Women's Business, and conducted in an appropriate way:

*Yeah, just yarning with all the women, you know, getting together, having a cuppa. I think someone just bringing it up, approaching it in a way where it's just a thing where you're sitting there having a yarn about–and not feeling shame.* P71

While some women felt as though they knew "enough" about cancer and cervical screening, others expressed a need for more information. Some women wanted to have a conversation with their doctor about screening, while others preferred to learn about screening in brochures and pamphlets available in the health centre waiting rooms, via TV advertisement, posters, or a smart phone application to avoid shame and awkward conversations with health professionals. These information channels were viewed as more "passive" and gentle methods of communication, bringing women's attention to the importance of screening without confrontation, embarrassment or "*preaching*" from health professionals. Still others reported a need for general awareness raising about cervical screening via community health promotion activities. Women suggested that women's health groups via social media and health services were a powerful and empowering resource in raising awareness about screening and starting

discussion about women's health issues. As mentioned above, women also discussed the importance of learning about women's health matters through family connections.

Women felt there was a higher level of awareness, discussion, general visibility and knowledge of breast cancer screening in the community compared to cervical screening. Women attributed the higher levels of breast screening awareness to several factors, including: the stigma associated with cervical screening taking place "down there"; that it was easier to use humour as a coping mechanism in relation to the breast screening procedure than the cervical screening test; and visual reminders of breast screening when the breast screening bus visited town, lamenting that there were no reminders that worked in the same manner for cervical screening. Women suggested that it was easier to complete breast screening because those clinics only provide one service, whereas cervical screening was just one of many other appointments to be scheduled at a PHCC.

Many women were unaware of the changes to the Australian cervical screening program, including the change in screening frequency from every two years to every five years. The women who were aware of the changes initially held reservations about the effectiveness of the new test given the longer interval between screens but were reassured by the increased sensitivity of the test. One woman, a health worker at an urban clinic, reported clients' positive reactions to the news of the change:

> . . . when I tell them it was five years, it's a five-year one, they're like, "Five years, even better," knowing that they were right for five years and it picks up early, detects things earlier, they were all for it. P01

Women also supported the change in terminology in the new program from "Pap smear" to "cervical screening", with one woman describing "smear" as a "*murder word*" (P35) as it kills conversations about screening.

## Relationship with the screening provider is critical

Some women wanted more culturally-appropriate screening services, including the availability of Aboriginal and Torres Strait Islander health professionals to conduct cervical screening. Women said they feel more comfortable with one "*of our own mob*" (P25) conducting cervical screening. Key strengths of this approach were the rapid development of strong, trusting relationships and feeling relaxed. Education about screening from Aboriginal Health Workers was also perceived to be particularly effective because women felt comfortable in their presence and could relate with them more easily than a doctor urging them to complete their screening. Sensitivity and cultural awareness from non-Indigenous health professionals was also important.

Most women viewed cervical screening as Women's Business, preferring that a female health professional conduct cervical screening. This view was particularly strong for women living in remote areas, who explained that screening was only discussed amongst other women including aunties, cousins and sisters, and only in women's spaces such as women's camps. Some women felt that screening should not be mentioned or advertised in areas where men may overhear. Women were comforted by the fact that female doctors had the same anatomy as them, and had been through the same experience:

> . . . they have got the body parts as me, you know what I mean? And if they've gone through something similar to this, they'll understand what it's like. . . . [We have] strong cultural values on Women's Business too. And it's an invasive procedure, that's how I see it. P52

Women reported that a key factor in commencing and continuing screening was a strong and trusting relationship with their General Practitioner (GP). Women built long-lasting relationships with their doctors, and felt that continuity of care, and the rapport that comes with it, was critical in helping them to feel comfortable screening. Women valued health professionals who took a holistic perspective of screening in the context of women's general health and life circumstances.

The way health professionals conveyed information (i.e., gentle and understanding) was just as important as the content of the information conveyed. For example, women had negative experiences when doctors forcefully "steamrolled" or hassled them about screening. Women reported feeling more comfortable with the screening test when doctors used plain language and could communicate clearly about what to expect during screening; educational pictures of the screening process also helped some women. Health professionals' clear and empathetic communication skills were particularly influential during women's first or early experiences of Pap smears and were important in helping women start and continue to screen.

Generally feeling comfortable and secure when visiting the PHCC reduced feelings of discomfort during screening:

> *Oh, when I have my testing done here they just make you feel so welcome and so warm, you don't feel so invaded, like a hospital. . .Yeah, you feel like a person, you don't seem–feel like a number.* P15

Women's perceptions of privacy, security, and confidentiality were important factors in decisions to participate in cervical screening. Decisions about whether to participate in screening were sometimes made based on features of the health service building, such as if the door could be locked, or whether there was a separate women's clinic.

## Logistical barriers to screening

Timing of appointments around work, family and other life commitments, transport, and the need to physically prepare, shaped decisions about screening. Women were time-poor and struggled to fit screening appointments in amongst commitments to work and family, which took higher priority in women's lives. The availability of screening services outside of work hours and the potential to reduce waiting times in the clinic by having a nurse available to complete screening rather than a doctor were raised as potential solutions. A participant who worked as a health worker in the clinic reported that many of her clients were more inclined to visit the clinic when transport to and from the clinic was provided. Women needed to physically prepare for screening by showering and feeling clean prior to the appointment, especially if visiting the PHCC after work.

Women also reported a range of logistical factors related to their experience of the cervical screening test. Women described the speculum as a "*duck beak*" or "*shoe horn*" and often noted how cold and uncomfortable it felt during the test. Some women described experiencing less discomfort if the health professional warmed the speculum before use. The size of the examination table was a concern for some women, who worried that they may not fit comfortably on the table due to having a larger body size. One woman with a physical disability stated the importance of an accessible, height-adjustable examination table that made screening more comfortable and easy for her.

## Issues for women who work in PHCCs

We yarned with 14 women employed in a range of roles at the PHCCs we visited, none of whom conducted cervical screening themselves. These women were able to provide their

perspectives both as women requiring cervical screening, and as health care professionals involved in facilitation of, or education about, cervical screening. They reported some unique issues: they felt uncomfortable being screened at the clinic by their colleagues and sometimes visited other clinics for screening to ensure privacy and confidentiality.

> *Especially after, you know, you finish your pap smear and then you're having lunch with them in the tea room, no thanks.* P28

Despite the barriers, women wanted to "lead by example" by keeping up with their own screening so that they could have more honest conversations with their clients about screening.

> *. . . and being a health worker too because then I can say, "No, I do mine," . . . it feels a bit shame but you can talk to your patient and encourage them to do it because you've done it and if I hadn't done it, well, I shouldn't be saying those things to the patient.* P25

A staff cervical screening day had been held in one PHCC shortly before yarns were conducted for the study. A female nurse from the local health district's women's health service visited the clinic and conducted cervical screening for all staff who wanted it. Women felt this was an excellent way to facilitate screening for women working in health services and other women, not only because it was convenient and promoted a sense of comradery, but also because the volume of screening afforded women a sense of anonymity:

> *. . . I think even if it wasn't a work thing and it was a group pap smear day type thing, at least you know it's just pap smear after pap smear after pap smear. So vagina, vagina, vagina, vagina, vagina. So she's [the health professional]not even going to think about your vagina by the time she's finished, the next person after you, because she's seen so many vaginas, do you know what I mean?* P49.

## Discussion

The views and experiences of Indigenous Australian women who participate in cervical screening are reflected in the themes of: screening as a means of staying strong and in control; overcoming fears, shame, and negative experiences of screening; the need to talk more openly about screening; relationships with screening providers are critical; logistical barriers to screenings; and issues for women employed at PHCCs.

Many of the findings were consistent with the factors previously reported that influence Indigenous women's cervical screening behaviour, such as knowledge and beliefs about cervical cancer, feelings of vulnerability and shame, pregnancy and first experiences of screening as key milestones in women's screening behaviour, health professionals' cultural awareness and sensitivity, the preference for female providers, the importance of having a strong and trusting relationships with both the health professional and PHCC, and logistical factors such as timing of appointments and transport [14–16]. Many of the findings were also consistent with the factors influencing cervical screening reported by Australian women in general [12, 13], the key difference being that Indigenous women's experience of these factors are compounded and heightened in the context of the ongoing impacts of colonisation, including intergenerational trauma, dispossession and culturally unsafe experiences in health care settings [29, 30].

A unique finding of this research was that Indigenous women sought control over their health through participating in cervical screening; as sovereign women they were driven by a

need for autonomy over their health. This finding is particularly important in the context of Indigenous Australians' disempowerment and ill health through the ongoing impacts of colonisation and government policy. Need for control over health was enacted in several ways, including the women actively participating in screening to stay healthy for themselves and their families, and through sharing knowledge about cervical screening and women's health matters with other women.

While many women spoke to the benefits of screening for their personal health, many women also perceived benefits to their families and communities. This aligns with Indigenous Australians' holistic and interconnected understandings of health and wellbeing [31, 32]; women's sense of being in control of their health and wellbeing was inherently linked with social relationships, kinship ties, and fulfillment of family and community roles, echoing previous findings [16]. Perceptions of the benefits of screening were held simultaneously with views of the discomfort involved in screening, like other women residing in countries with established screening programs [11]. Despite these views, women were able to overcome the emotional obstacles to participate in screening.

The findings demonstrate how feelings of control and empowerment may support Indigenous women's participation in screening. The implication of these findings is that enhanced control may increase women's willingness to participate in cervical screening, suggesting that health professionals should explore the option of HPV self-collection with women when discussing cervical screening. This option affords women choice and agency over how cervical screening is conducted. Māori women, who frequently reported a need to retain control over their bodies during cervical screening, indicated a preference for HPV self-collection over a clinician-collected sample, providing evidence for the potential for self-collection to restore a sense of control [17].

Women employed in PHCCs wanted to be role models for their clients by participating in cervical screening and yet faced unique challenges in relation to privacy and confidentiality. Cervical screening conducted by an external screening provider was one successful strategy that overcame these barriers. This finding is particularly important given that 2016 Census data indicates that health care and social assistance was the most common employment industry for Indigenous women [33]. It is critical that strategies for participation in cervical screening consider the needs and concerns of this large workforce of Indigenous women.

There is much to be learned from women's views and experiences. We have developed a list of recommendations for health professionals and PHCCs (Table 2).

## Strengths and limitations

The strength of this research was the study design which privileged Indigenous women's voices in multiple ways: through their participation in the study, leadership of the investigatory team and researchers, and involvement in the analysis. This approach is consistent with Rigney's Indigenist research approach [34]. Our approach to knowledge creation upholds Indigenous Australians' right to control and autonomy over health and wellbeing outcomes in the context of colonisation and its ongoing effects. Such an approach is increasingly preferred among other Indigenous populations; for instance, Kaupapa Māori research (research which challenges the dominant Western discourse and power by privileging Maori worldviews) is guided by the principle of *tino Rangatiratanga*, translating to Māori sovereignty, self-determination, and autonomy [35–37]. Indigenist and Kaupapa Māori research approaches centre Indigenous peoples' views, experiences, and knowledge systems in research methodology, ensuring that outcomes are relevant and beneficial.

**Table 2. Authors' recommendations for health professionals and Primary Health Care Clinics based on screened Indigenous women's views and experiences of cervical screening.**

| Theme | Recommendation [a] |
|---|---|
| Enhancing strength and control | • Frame discussions about cervical screening and general health in terms of how women can take control of their health.<br>• Emphasise benefits of cervical screening for the women personally, but also for family and community, through living long and healthy lives<br>• Avoid placing too much pressure on women to screen opportunistically<br>• Emphasise the short duration of cervical screening as a positive feature<br>• Emphasise that cervical screening is part of a holistic women's health routine<br>• Move conversations beyond that screening is necessary; explain why screening is beneficial.<br>• Discuss HPV self-collection with eligible women as a means to enhance control over health |
| Overcoming fears, shame, and negative experiences. | • Acknowledge emotional obstacles to screening, including shame, discomfort and feelings of invasiveness<br>• Ensure results are rapidly and clearly communicated back to women to avoid anxiety<br>• Maternal health checks are a good opportunity to discuss screening with women, as they feel less shame about women's health checks after the experience of pregnancy and childbirth<br>• Acknowledge that a woman's first experience of cervical screening can shape her attitudes to cervical screening in the future. Ensuring that the first screen is a positive experience may support women to continue screening into the future. |
| Talking more openly about screening | • Consider women's differing information needs. Check with women to ensure the amount of information provided is enough to make them feel comfortable screening<br>• Provide women with a variety of information channels including both "active" (e.g., discussion with health professional) and "passive" (e.g., brochure in waiting room) options<br>• Use social media, where appropriate in the community and using appropriate language, to promote women's health awareness and events in the clinic<br>• Facilitate women's group meetings at the PHCC to encourage women to talk with health professionals and with other women about screening and other health matters in approachable and comfortable setting<br>• Encourage women to share knowledge gained with other women; community champions may be helpful<br>• Explain changes to the cervical screening program, emphasising changes to terminology, longer screening intervals, and increased sensitivity of the test, as women responded positively to these aspects |
| Enhancing screening providers' relationships with Indigenous women | • Ensure that all staff practice culturally safe health care delivery<br>• Recognise the importance of building long-term, trusting relationships with women<br>• Employ Aboriginal and Torres Strait Islander staff<br>• Ensure there are female health professionals available to conduct cervical screening; confirm women's preference for who conducts screening.<br>• Recognise that cervical screening is generally viewed as Women's Business; this means that women may not wish to discuss cervical screening with male professionals or see cervical screening information displayed in shared men's and women's spaces in the clinic<br>• Ensure appropriate measures to assure women of their privacy and confidentiality when screening are put in place (e.g., separate women's clinic or private and discrete clinical rooms)<br>• Explain cervical screening and its benefits in a gentle and empathetic manner, using clear and simple language. Avoid medical jargon.<br>• Acknowledge the importance of health professionals' trusting and long-lasting relationships as a key factor in helping women to participate in screening |
| Overcoming logistical barriers to screening | • Implement flexible appointment scheduling in the clinic, such as extended business hours so that women may attend before or after work<br>• Ensure there are a variety of staff available (such as nurses) to conduct screening to avoid longer waiting times for doctors<br>• Connect women with transport options to attend the clinic<br>• Provide space and time for women to physically prepare to cervical screening, such as the option to shower or freshen up<br>• Warm the speculum before conducting screening<br>• Ensure examination tables and clinical spaces meet women's physical needs, including women of different body sizes and with disability. |
| Supporting women employed in PHCCs to screen | • Support female staff to attend health appointments, including cervical screening, with flexible working options<br>• Ensure female staff's right to privacy and confidentiality are respected in the clinic<br>• Provide opportunities for female staff to complete cervical screening during work hours<br>• Conduct a staff screening day. Organise for a health professional external to the PHCC to conduct the screening to enhance female staff member's sense of privacy and anonymity. |

[a] Recommendations were developed by the authors based on the outcomes of the qualitative analysis and were not validated by participants.

The NCSP was in a transition period to the renewed program at the time the data was collected for this study. Therefore, we acknowledge that the criteria used in this study to define women as "screened" was inconsistent with Australian guidelines at the time. The criteria of having screened within five years was chosen because under the renewed NCSP, asymptomatic women would be up-to-date with screening if they had screened within the last five years [38]. Under the transition guidelines, a woman who hadn't had a Pap test in the last two years would technically be overdue for cervical screening [39]. This means that some of the women in the "screened" sample were overdue for screening under the transition guidelines.

There are many Indigenous Australian nations, each with diverse cultures, and as such this study cannot represent all Indigenous Australian women's views on cervical screening. However, the study considered the views and experiences of women from different geographic regions. The findings may have relevance to other populations, including other global Indigenous populations or minority groups with reduced cervical screening rates.

While some findings may be used to develop strategies to support Indigenous women who do not participate in cervical screening to start screening, it is important to note that not all findings may be appropriate for this group of women. These women require tailored strategies informed by their views and experiences to support their participation in cervical screening. A final limitation is that, due to logistical and financial restrictions, the findings of the qualitative analysis and recommendations were not validated with the participants before preparation for publication. To bolster the validity of the findings, the analysis was led by an Aboriginal woman (TB), and the findings were reviewed by Aboriginal and Torres Strait Islander women (LW and GG) with expertise in cancer and cancer prevention.

## Conclusions

The findings of this study indicate that cervical screening should be conducted in a way that empowers Indigenous women to feel in control of their bodies, health, and health decision-making. While women acknowledged that cervical screening was an uncomfortable process, women overcame the emotional obstacles by recognising the benefits for themselves, their families, and their communities. The findings may also inform future public health campaigns and other cancer screening programs, such as breast screening and bowel screening. Ultimately, prioritising sovereign Indigenous women's views and rights to control their health may support women to start and maintain participation in cervical screening, and ensure that Australia's cervical cancer elimination goal includes all Australian women.

## Supporting information

**S1 Text. Modified Screening Matters interview guide.**
(DOCX)

## Acknowledgments

The views expressed in this publication are those of the authors and do not necessarily reflect the views of the funder. We wish to thank the women who yarned with us for this research. We also thank the Aboriginal women who conducted the yarns: two community research officers in Central Australia (who cannot be identified to ensure anonymity of the participating sites) and Ms Beverley Marcusson (BM). Finally, we are grateful for the contributions of staff in the Central Australian site and a second New South Wales site to the *Screening Matters* study. Ownership of Aboriginal and Torres Strait Islander knowledge and cultural heritage is retained by the informant.

## Author Contributions

**Conceptualization:** Tamara L. Butler, Kate Anderson, John R. Condon, Gail Garvey, Julia M. L. Brotherton, Joan Cunningham, Allison Tong, Suzanne P. Moore, Lisa J. Whop.

**Formal analysis:** Tamara L. Butler, Kate Anderson.

**Funding acquisition:** Kate Anderson, John R. Condon, Gail Garvey, Julia M. L. Brotherton, Joan Cunningham, Allison Tong, Suzanne P. Moore, Lisa J. Whop.

**Investigation:** Tamara L. Butler, Lisa J. Whop.

**Methodology:** Tamara L. Butler, Kate Anderson, John R. Condon, Gail Garvey, Julia M. L. Brotherton, Joan Cunningham, Allison Tong, Suzanne P. Moore, Lisa J. Whop.

**Project administration:** Tamara L. Butler, Clare M. Maher, Jacqueline K. Mein, Eloise F. Warren, Lisa J. Whop.

**Resources:** Clare M. Maher, Jacqueline K. Mein, Eloise F. Warren.

**Supervision:** Lisa J. Whop.

**Visualization:** Tamara L. Butler.

**Writing – original draft:** Tamara L. Butler, Lisa J. Whop.

**Writing – review & editing:** Tamara L. Butler, Kate Anderson, John R. Condon, Gail Garvey, Julia M. L. Brotherton, Joan Cunningham, Allison Tong, Suzanne P. Moore, Clare M. Maher, Jacqueline K. Mein, Eloise F. Warren, Lisa J. Whop.

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
