## [Decision Letter · Decision Letter 0]

26 Feb 2020

PONE-D-20-01555

Indigenous Australian women's experiences of participation in cervical screening

PLOS ONE

Dear Dr. Butler: 

Thank you for submitting your manuscript to PLOS ONE. After careful consideration, we feel that it has merit but does not fully meet PLOS ONE’s publication criteria as it currently stands. Therefore, we invite you to submit a revised version of the manuscript that addresses the points raised during the review process.

Please note that we feel that all of the reviewers comments require your attention. Additional comments have also been added by the Editor. 

We would appreciate receiving your revised manuscript by April 24, 2020. To enhance the reproducibility of your results, we recommend that if applicable you deposit your laboratory protocols in protocols.io, where a protocol can be assigned its own identifier (DOI) such that it can be cited independently in the future. For instructions see: http://journals.plos.org/plosone/s/submission-guidelines#loc-laboratory-protocols

We look forward to receiving your revised manuscript.

Kind regards,

Nelly Oelke

Academic Editor

PLOS ONE

Additional Editor Comments (if provided):

Page 8: Lines 157-158: It is no clear who did the other interviews in other PHCCs in the data collection section.

Page 8: Lines 160-162: The last two sentences in the recruitment section need to be moved to data collection.

Page 8: Lines 164-165: How were Indigenous women involved and how did they lead the study?

Page 9: Line 173: Suggest removing the reference to Table 1 here as it is not the survey and move Table 1 to the results.

Page 11: Line 200: Please add the version of NVivo.

Page 12: Results section – in identifying the age group of the woman in the quote, could this be potentially identifiable, particularly those that are employed at PHCCs? And I am not sure it adds a lot.

Page 25: Lines 549-551: The table and text regarding recommendations should be part of the results, not including the actual data in the conclusion or discussion.

Journal Requirements:

We note that one or more of the authors are employed by a commercial company: Wuchopperen Health Service Pty Ltd.

Reviewers' comments:

Reviewer's Responses to Questions

**Comments to the Author**

1. Is the manuscript technically sound, and do the data support the conclusions?

Reviewer #1: Partly

Reviewer #2: Yes

2. Has the statistical analysis been performed appropriately and rigorously? 

Reviewer #1: N/A

Reviewer #2: N/A

3. Have the authors made all data underlying the findings in their manuscript fully available?

Reviewer #1: Yes

Reviewer #2: Yes

4. Is the manuscript presented in an intelligible fashion and written in standard English?

Reviewer #1: Yes

Reviewer #2: Yes

5. Review Comments to the Author

Reviewer #1: A clear articulation and analysis using a colonisation/power framework would add real strength to the information being reported in this paper. The majority of barriers to Indigenous women’s participation in cervical screening are system levels factors and so a more robust and nuanced discussion of the significance of these barriers as they relate to this study would greatly add to the paper overall. Five of the 6 key themes identified from the findings relate to system ‘failures’ but are not discussed as such. The role of system-level factors in perpetuating unmet need, discrimination and health service racism have all been extensively documented in other studies and are key areas which should be fully elaborated on as relevant context for these findings.

In other places, the paper would be better served if again, the text was re-written to reflect system failures, for example, both the abstract and the introduction note ‘Many Indigenous Australian women do not regularly participate in cervical screening which contributes to high cervical cancer incidence and mortality.’ This implies that Indigenous women are at fault when in fact it is the screening programme which has not served them as evidenced by high cervical cancer incidence and mortality rates.

Table 2 should be inserted and referred to at the beginning of the ‘Discussion’ section not put in as part of the conclusion.

Reviewer #2: REVIEWER COMMENTS

----------------

The manuscript focuses on 50 Indigenous Australian women who have undergone cervical screening within the last five years from five PHCCs. The researchers have chosen a culturally-appropriate information gathering (i.e. data collection) method of yarning to identify the main themes focused around cervical screening. Recommendations for healthcare providers were also provided.

----------------

Biggest Concern

The researchers have used a culturally-appropriate research method through `yarning` to empower the voices of the Indigenous women. To continue culturally-appropriate engagement with the Indigenous women:

1. Were the final six themes or recommendations (Table 2) validated by the participating women?

2. How were the women involved in the `analysis` or decision-making process for consensus of the final themes and recommendations? If not, are there plans to engage the women to provide them with the findings or to include them in the finalization of recommendations?

Please make sure to address how the participants were engaged throughout the research process. If there was no engagement past the `yarning`, please include this within the limitations.

----------------

Participants and Recruitment

Lines 157 – 162 should be included in Data Collection as it is not part of Participants and Recruitment.

Researchers state that initially the aim was to yarn with 10 women at each participating PHCC; however, they state that more women than expected were recruited. It also mentions that data saturation was achieved within each PHCC (Lines 158 – 161).

1. How many women were included (i.e. yarned with) from each PHCC?

2. What is the meaning of data saturation being achieved within each PHCC?

3. Were the participants representative across the five PHCCs?

Researchers also mention that in one PHCC, trained staff members conducted yarns. Who interviewed the women at the other four PHCCs? Why were staff members from the other PHCCs not trained to conduct the yarns?

As well, Lines 184 – 188; there is no mention that staff of the one PHCC were included in conducting the yarns. There is mention that there were pre-existing relationships between two research officers. Were there pre-existing relationships with the women and staff of PHCC?

----------------

Data Collection

Table 1 includes both demographic and health survey data. The results discuss HPV vaccination (Lines 227 – 229); however, this is not included in Table 1.

1. Are the health survey data included Table 1? It seems that only the demographic data has been included.

2. I would suggest that Table 1 be moved to the Results discussion. As well, if there are missing data or data not shown in the manuscript (i.e. HPV vaccination) to include this within Table 1 or a separate table or mention that the data is not shown.

Please ensure that all data collected and discussed has been included within the manuscript.

----------------

Conclusions

The recommendations (Table 2) should be introduced as part of the Results section (not the Conclusions section).

1. Please include who was involved in the consensus or decision-making process for the final recommendations provided.

----------------

Minor Changes

Line 56: the incidence has decreased by more than half. Please change the wording.

Line 82: Change of to for

Line 82 – 83: ``a similar pattern …``: this sentence is unclear.

Line 139: Suggest to change Method to RESEARCH METHODS

Line 145; Line 223: Add reference for the Screening Matters project

Line 230: Suggest to clarify that the themes are identified from analysis of the yarns

Line 325: Use double quotation marks to keep consistency throughout the manuscript - “down there”

Line 384: Italicize or use quotation for Women’s Business instead of capitalizing.

Line 396: Use the full term for GP (although a common abbreviation, abbreviations should be introduced prior to use).

6. PLOS authors have the option to publish the peer review history of their article (what does this mean?). If published, this will include your full peer review and any attached files.

Reviewer #1: No

Reviewer #2: No

---

## [Author Response · Author response to Decision Letter 0]

5 May 2020

12 April 2020

Nelly Oelke

Academic Editor 

PLOS ONE 

Re: Article ID PONE-D-20-01555

Title: Indigenous Australian women's experiences of participation in cervical screening

Dear Nelly Oelke,

Thank you for your and the reviewers’ comments on our manuscript. We welcome the review and feel that the manuscript has improved as a result of addressing the comments. 

In the letter below, we outline our response to the issues raised in your email of 27th February 2020. For clarity, we have numbered each issue raised and provide our response underneath. Line references refer to the marked-up manuscript. In the revised manuscript, revisions are indicated using the Track Changes function of Microsoft word using the “show all revisions inline” setting.

Thank you again for your feedback and I look forward to hearing from you again in due course. 

Yours sincerely,

Dr Tamara Butler on behalf of the authors of PONE-D-20-01555

Additional Editor Comments (if provided):

1. Page 8: Lines 157-158: It is no clear who did the other interviews in other PHCCs in the data collection section.

To clarify this we have rearranged information in the Data Collection section and added some text. Please see lines 221-229. 

2. Page 8: Lines 160-162: The last two sentences in the recruitment section need to be moved to data collection.

Revised as requested. See lines 254-255. The information about data saturation in clarified in lines 256-260.

3. Page 8: Lines 164-165: How were Indigenous women involved and how did they lead the study?

We have created a new section in the Research methods section (“The Screening Matters study approach”, lines 164-180) to clarify this statement. 

4. Page 9: Line 173: Suggest removing the reference to Table 1 here as it is not the survey and move Table 1 to 298-299.

5. Page 11: Line 200: Please add the version of NVivo.

Revised as requested. Please see line 266.

6. Page 12: Results section – in identifying the age group of the woman in the quote, could this be potentially identifiable, particularly those that are employed at PHCCs? And I am not sure it adds a lot.

Revised as requested throughout the Results section (from line 289). The age groups have been removed.

7. Page 25: Lines 549-551: The table and text regarding recommendations should be part of the results, not including the actual data in the conclusion or discussion.

Reviewer 1 and 2 also noted placement of Table 2. Reviewer 1 suggested moving the table to the Discussion section and Reviewer 2 suggested moving the Table to the Results section. As the recommendations were not an aim of the primary qualitative analysis and were developed as a secondary outcome after analysis of the yarns had been completed, it has been placed in the Discussion section. We have added a legend to Table 2 and revised the title of Table 2 to ensure this is clear. See line 543-546.

Journal requirements

8. Please ensure that your manuscript meets PLOS ONE's style requirements, including those for file naming. The PLOS ONE style templates can be found at http://www.plosone.org/attachments/PLOSOne_formatting_sample_main_body.pdf and http://www.plosone.org/attachments/PLOSOne_formatting_sample_title_authors_affiliations.pdf

Several changes have been made throughout the manuscript to ensure it meets style requirements, including changing table formatting, formatting of the affiliation page, formatting of the manuscript main body, and supporting information file name. If there is a specific error in style we would be happy to correct it at your instruction.

Please note that Julia Brotherton’s affiliation “VCS Foundation” is correct in its abbreviated format on the affiliation page (line 16).

9. Thank you for stating the following in the Competing Interests section: "The authors have declared that no competing interests exist." 

We note that one or more of the authors are employed by a commercial company: Wuchopperen Health Service Pty Ltd. 

A. Please provide an amended Funding Statement declaring this commercial affiliation, as well as a statement regarding the Role of Funders in your study. If the funding organization did not play a role in the study design, data collection and analysis, decision to publish, or preparation of the manuscript and only provided financial support in the form of authors' salaries and/or research materials, please review your statements relating to the author contributions, and ensure you have specifically and accurately indicated the role(s) that these authors had in your study. You can update author roles in the Author Contributions section of the online submission form. 

Please also include the following statement within your amended Funding Statement. “The funder provided support in the form of salaries for authors [insert relevant initials], but did not have any additional role in the study design, data collection and analysis, decision to publish, or preparation of the manuscript. The specific roles of these authors are articulated in the ‘author contributions’ section.” 

Changes to the Competing Interest Statement are highlighted in yellow below. Please note that the commercial entities (Southern Queensland Centre of Excellence in Aboriginal and Torres Strait Islander Primary Health Care, Wuchopperen Health Service Pty Ltd and Yerin Eleanor Duncan Aboriginal Health Centre) did not fund the study, only provided salary support for three authors and authorised access to participants. 

Financial Disclosure

This study was supported by the National Health and Medical Research Council (NHMRC) funded Centre of Research Excellence (CRE) in Targeted Approaches To Improve Cancer Services for Aboriginal and Torres Strait Islander Australians (TACTICS; #1153027), the NHMRC-funded CRE in Discovering Indigenous Strategies to improve Cancer Outcomes Via Engagement, Research Translation and Training (DISCOVER-TT; #1041111), and the Cancer Council NSW Strategic Research Partnership to improve cancer control for Indigenous Australians (STREP Ca-CIndA; SRP 13-01, with supplementary funding from Cancer Council WA). 

The study is affiliated with the following commercial entities: Southern Queensland Centre of Excellence in Aboriginal and Torres Strait Islander Primary Health Care, Wuchopperen Health Service Pty Ltd and Yerin Eleanor Duncan Aboriginal Health Centre. These commercial affiliations provided support in the form of salaries for authors CM, JM, and EW, respectively, and provided permission to collect data from clients, but did not have any additional role in the study design, and data analysis, decision to publish, or preparation of the manuscript. The specific roles of these authors are articulated in the ‘author contributions’ section.

TB was supported by an Australian Research Council Discovery Australian Aboriginal and Torres Strait Islander Award (#IN190100050). GG and LJW were funded by National Health and Medical Research (NHMRC) Early Career Fellowships (#1105399 and #1142035, respectively). JCu was funded by an NHMRC Research Fellowship (#1058244). AT was supported by a NHMRC Career Development Fellowship (#1106716). KA, JCo, JB, SM, JM, EW, and CM received no specific funding for this work.

The views expressed in this publication are those of the authors and do not necessarily reflect the views of the funders. The funders had no role in study design, data collection and analysis, decision to publish, or preparation of the manuscript. 

B. Please also provide an updated Competing Interests Statement declaring this commercial affiliation along with any other relevant declarations relating to employment, consultancy, patents, products in development, or marketed products, etc. 

Changes to the Competing Interest Statement are highlighted in yellow below. Please note that while the commercial entities do not restrict access to data and materials, we are still bound by ethical obligations to avoid individuals being identified in qualitative data. As such we have referred to the Data Availability statement for the restrictions in place. 

Competing Interests

The authors have declared that no competing interests exist. The commercial affiliations do not alter our adherence to PLOS ONE policies on sharing data and materials. Restrictions on sharing data and materials relate to ethical reasons only (see Data Availability statement).

Reviewer 1

10. Reviewer #1: A clear articulation and analysis using a colonisation/power framework would add real strength to the information being reported in this paper. The majority of barriers to Indigenous women’s participation in cervical screening are system levels factors and so a more robust and nuanced discussion of the significance of these barriers as they relate to this study would greatly add to the paper overall. Five of the 6 key themes identified from the findings relate to system ‘failures’ but are not discussed as such. The role of system-level factors in perpetuating unmet need, discrimination and health service racism have all been extensively documented in other studies and are key areas which should be fully elaborated on as relevant context for these findings.

Reviewer 1 suggests that an analysis using a colonisation/power framework would add strength to the paper. While we acknowledge the reasoning for this suggestion, for this approach to have scientific rigour and integrity, it would have been ideal to identify the framework or model at the conceptualisation of the study and design the study around this. While we broadly aimed to investigate individual, community and structural barriers and facilitators to cervical screening, we did not use a specific framework for this investigation. Re-analysing the data post-hoc using a different framework would not be appropriate at this stage. 

Our approach to the study aimed to privilege the voices of Indigenous Australian women, and we feel this is strongly aligned with the principles of a power/colonisation framework. We have done this in several ways including the through the study leadership, women conducting and participating in the yarns, the research methodology, and Indigenous leadership in analysis. We have created a new section in the Research methods section (“The Screening Matters study approach”, lines 164-180) emphasise our approach to knowledge creation. We also outline this as a strength of the paper in lines 593-605. Our approach is aligned with Rigney’s Indigenist research approach, which also aligns with the philosophies of a power/colonisation framework. Indigenist research is: “research by Indigenous people whose primary informants are Indigenous people and whose goals are to serve and inform the Indigenous struggle for self-determination.” (Rigney, 1999, p. 118). 

We provide the relevant context for this study, including factors such as racism, a lack of cultural awareness and sensitivity among health professionals, in the Introduction section (Lines 120-138). However, we recognise the framing of these barriers could have more strongly emphasised the structural barriers to Indigenous women participating in screening. We have made a number of changes and clarifications within this section (lines 120-138) to more strongly emphasise this point (also see item 11 below). 

11. In other places, the paper would be better served if again, the text was re-written to reflect system failures, for example, both the abstract and the introduction note ‘Many Indigenous Australian women do not regularly participate in cervical screening which contributes to high cervical cancer incidence and mortality.’ This implies that Indigenous women are at fault when in fact it is the screening programme which has not served them as evidenced by high cervical cancer incidence and mortality rates. 

We have carefully revised several sections of the paper to ensure that it reflects or incorporates structural failures, including:

• Lines 35-39 in the Abstract have been revised to note that the National Cervical Screening Program is failing to meet the needs of Indigenous Australian women. (Please also note this meant that the entire Abstract needed to be edited to ensure it was no longer than 300 words as per journal guidelines). 

• In the Introduction lines 84-86 we state that cervical screening program does not meet the needs of Indigenous women. 

• Lines 120-138 have been clarified to ensure barriers incorporate structural as well as individual barriers (supported by evidence). 

• Lines 555-558 places the barriers shared with non-Indigenous women in the context of ongoing effects of colonisation and barriers to health care access, thus heightening and compounding the impact on Indigenous women. 

There are numerous other small adjustments throughout the Introduction section to better reflect structural barriers. 

12. Table 2 should be inserted and referred to at the beginning of the ‘Discussion’ section not put in as part of the conclusion.

Revised as requested. See line 543-546.

Reviewer 2 

13. Biggest Concern. The researchers have used a culturally-appropriate research method through `yarning` to empower the voices of the Indigenous women. To continue culturally-appropriate engagement with the Indigenous women:

A. Were the final six themes or recommendations (Table 2) validated by the participating women? 

These recommendations were developed by the authors based on the outcomes of the qualitative analysis and were not validated by participants. We have added a legend to Table 2 (lines 543-546) to clarify this. For this reason, we have not reported Table 2 in the Results section and instead moved to the Discussion section. 

B. How were the women involved in the `analysis` or decision-making process for consensus of the final themes and recommendations? If not, are there plans to engage the women to provide them with the findings or to include them in the finalization of recommendations?

Please make sure to address how the participants were engaged throughout the research process. If there was no engagement past the `yarning`, please include this within the limitations.

The themes presented in the Results and the table of recommendations were not validated by participating women. We were not able to do this due to logistical and financial barriers to re-interviewing women who participated in the study and travelling to study sites a second time. We have clarified that the recommendations were not validated by participating women in the legend of Table 2, just above line 546, as stated in item 13A. We have also clarified that themes were not validated by participants in line 278-279.

As suggested by Reviewer 2, we also note this as a limitation in lines 624-628, noting that the validity of the findings is supported by the fact that the analysis was conducted by an Aboriginal woman (TB), and two other Aboriginal and Torres Strait Islander women (LW and GG) with professional expertise in cancer and cancer prevention reviewed the findings to ensure their validity.

Participants and Recruitment

14. Lines 157 – 162 should be included in Data Collection as it is not part of Participants and Recruitment.

This has been removed from Participants and Recruitment and incorporated into the Data Collection section (lines 221-229).

Please note that several changes have been made throughout the Research methods section to clarify meaning and ensure information is presented in a logical order. 

15. Researchers state that initially the aim was to yarn with 10 women at each participating PHCC; however, they state that more women than expected were recruited. It also mentions that data saturation was achieved within each PHCC (Lines 158 – 161).

A. How many women were included (i.e. yarned with) from each PHCC?

We have included further data in Table 1 to clarify the number of women we yarned with at each PHCC (see the last 7 lines of Table 1). We have also included some clarification and a range of women recruited per PHCC (see line 201-202) 

B. What is the meaning of data saturation being achieved within each PHCC?

Our approach to data saturation has been clarified in lines 256-260.

C. Were the participants representative across the five PHCCs?

While we aimed to yarn with a diverse range of women of different ages and geographic region, obtaining a representative sample was not a focus of recruitment. We have clarified that women were recruited by convenience sampling in line 191.

16. Researchers also mention that in one PHCC, trained staff members conducted yarns. Who interviewed the women at the other four PHCCs? Why were staff members from the other PHCCs not trained to conduct the yarns?

This has been clarified in lines 221-229.

17. As well, Lines 184 – 188; there is no mention that staff of the one PHCC were included in conducting the yarns. There is mention that there were pre-existing relationships between two research officers. Were there pre-existing relationships with the women and staff of PHCC?

The relationship between the research officers and the women (participants) has been clarified in lines 240-245. The context added to lines 221-229 should also serve to clarify the role of the research officers. 

Data Collection

18. Table 1 includes both demographic and health survey data. The results discuss HPV vaccination (Lines 227 – 229); however, this is not included in Table 1.

A. Are the health survey data included Table 1? It seems that only the demographic data has been included.

Yes, health survey data is included. This relates to data regarding presence of one or more chronic disease. 

We have now updated Table 1 to include the whether women reported having had the HPV vaccination. These data were not originally reported in Table 1 as many women indicated that they were unsure of their answer. We have added a legend to Table 1 indicating this. See above line 229.

19. I would suggest that Table 1 be moved to the Results discussion. As well, if there are missing data or data not shown in the manuscript (i.e. HPV vaccination) to include this within Table 1 or a separate table or mention that the data is not shown. Please ensure that all data collected and discussed has been included within the manuscript.

Table 1 has been moved to the Results section. See Line 298. We have addressed the HPV vaccination data issue in 18A.

Conclusions

20. The recommendations (Table 2) should be introduced as part of the Results section (not the Conclusions section).

After considering the reviews from the Editor and both Reviewers on the position of Table 2, we have decided to move Table 2 to the start of the Discussion section. The recommendations were developed by the authors based on the outcomes of the qualitative analysis and so we do not wish to represent them as findings of the qualitative analysis but rather our recommendations based on the findings. 

A. Please include who was involved in the consensus or decision-making process for the final recommendations provided.

This has been addressed in 13A and 13B above.

Minor Changes

21. Line 56: the incidence has decreased by more than half. Please change the wording.

Revised as requested. See line 68.

22. Line 82: Change of to for

Revised as requested. See line 95.

23. Line 82 – 83: ``a similar pattern …``: this sentence is unclear.

This sentence has been revised. See lines 96-98.

24. Line 139: Suggest to change Method to RESEARCH METHODS

Revised as requested. See line 161.

25. Line 145; Line 223: Add reference for the Screening Matters project

As this is the first publication for the project, there is no reference for the Screening Matters project. The Protocol was not published. 

26. Line 230: Suggest to clarify that the themes are identified from analysis of the yarns

Revised as requested. See line 300. 

27. Line 325: Use double quotation marks to keep consistency throughout the manuscript - “down there”

In reviewing the manuscript, we have changed the title of this theme to “Needing to talk more openly about screening”, so double quotation marks are no longer required. See line 584. The change to the title has been carried throughout the manuscript. 

28. Line 384: Italicize or use quotation for Women’s Business instead of capitalizing.

To denote this as a specific cultural practice we would prefer to captialise the term Women’s Business. Quotations and italics portray the phrase as “other” and exotic, and we seek to centre Indigenous women’s views in our approach. 

29. Line 396: Use the full term for GP (although a common abbreviation, abbreviations should be introduced prior to use).

Revised as requested. See line 467.

---

## [Editor Report · Decision Letter 1]

18 May 2020

PONE-D-20-01555R1

Indigenous Australian women's experiences of participation in cervical screening

PLOS ONE

Dear Dr. Butler:

Thank you for submitting your manuscript to PLOS ONE. After careful consideration, we feel that it has merit but does not fully meet PLOS ONE’s publication criteria as it currently stands. Therefore, we invite you to submit a revised version of the manuscript that addresses the points raised during the review process.

Please see notes in the review section. All comments must be addressed. 

We would appreciate receiving your revised manuscript by July 14, 2020. To enhance the reproducibility of your results, we recommend that if applicable you deposit your laboratory protocols in protocols.io, where a protocol can be assigned its own identifier (DOI) such that it can be cited independently in the future. For instructions see: http://journals.plos.org/plosone/s/submission-guidelines#loc-laboratory-protocols

We look forward to receiving your revised manuscript.

Kind regards,

Nelly Oelke

Academic Editor

PLOS ONE

Additional Editor Comments (if provided):

Thank you for submitting your revised paper. For the most part, revisions have addressed the concerns of reviewers. There still seem to be a few outstanding issues to be addressed along with a few minor edits required.

1. On page 9, Line 177, "understanding" should be "understand"

2. On page 10, Line 199, you state that you recruited both screened and unscreened women to participate. But the study reports only screened women were recruited or at the least that this data set only includes data from screened women. This needs to be clarified at the very least in the number of places where you differentiate between the two. But more so, if you have data from non-screened women, should these voices not also be reported?

3. On page 18, Line 314, "to be" should be changed to "for the."

4. With regards to Table 2, I think it should be in the discussion as you suggest, but think that it would be better placed after the discussion of your analyzed data. Discussions are generally written with the discussion of the data first, and then provide recommendations.

Reviewers' comments:

N/A

---

## [Author Response · Author response to Decision Letter 1]

22 May 2020

19 May 2020

Nelly Oelke

Academic Editor 

PLOS ONE 

Re: Article ID PONE-D-20-01555R1

Title: Indigenous Australian women's experiences of participation in cervical screening

Dear Nelly Oelke,

Thank you for comments on the revised manuscript raised in your email of 19 May 2020. Below we respond to your comments. Line references refer to the marked-up manuscript. In the revised manuscript, revisions are indicated using the Track Changes function of Microsoft word using the “show all revisions inline” setting.

I apologise that the manuscript and attachments are out of order in the submission. The system did not allow me to reorder them (I tried the drag and drop method and manual number re-ordering, and two different web browsers). They are labelled correctly. I apologise for the inconvenience. 

Yours sincerely,

Dr Tamara Butler on behalf of the authors of PONE-D-20-01555R1

Additional Editor Comments (if provided):

1. On page 9, Line 177, "understanding" should be "understand".

Revised as requested. See page 7, line 171. I also corrected an error in punctuation on line 170.

2. On page 10, Line 199, you state that you recruited both screened and unscreened women to participate. But the study reports only screened women were recruited or at the least that this data set only includes data from screened women. This needs to be clarified at the very least in the number of places where you differentiate between the two. But more so, if you have data from non-screened women, should these voices not also be reported?

To clarify this matter, I have moved the reference to unscreened women to page 7, line 163-165. This contextualises the recruitment of both screened and unscreened women in the larger Screening Matters study. I have also clarified terminology in this section by replacing “unscreened” women with “under-screened and never-screened” women in lines 160 and 164.

I have also added emphasis on the fact that only screened women’s views are reported in this paper on page 7, line 166-167, and this is also highlighted under Participants and Recruitment on page 8, line 176-177. 

I searched for other references to unscreened women in the paper and could not find further places where unscreened women had been referenced, however I would be happy to clarify any other areas at your further instruction. 

We decided to report the views of screened and under-screened/never-screened women’s views separately because strategies to enhance women’s participation in cervical screening differ greatly between those women who are already screening regularly and those who do not screen regularly. Furthermore, during the yarns we found that under-screened/never-screened women reported several issues that need to be reported with sensitivity and care (e.g., experiences of sexual trauma and discriminatory health care provision). Under-screened/never-screened women’s voices are rarely heard and are not well represented in the literature. We felt we could do the voices of both groups of women justice by separately reporting the experiences of each group. The manuscript for under-screened/never-screened women is currently in development. 

3. On page 18, Line 314, "to be" should be changed to "for the."

Revised as requested. See page 14, line 274.

4. With regards to Table 2, I think it should be in the discussion as you suggest, but think that it would be better placed after the discussion of your analysed data. Discussions are generally written with the discussion of the data first, and then provide recommendations.

Revised as requested. Table 2 and text introducing the recommendations have been moved to after the discussion of findings. See page 29, line 550- page 33, line 555.

---

## [Editor Report · Decision Letter 2]

29 May 2020

Indigenous Australian women's experiences of participation in cervical screening

PONE-D-20-01555R2

Dear Dr. Butler,

We are pleased to inform you that your manuscript has been judged scientifically suitable for publication and will be formally accepted for publication once it complies with all outstanding technical requirements.

With kind regards,

Nelly Oelke

Academic Editor

PLOS ONE
---

## [Editor Report · Acceptance letter]

5 Jun 2020

PONE-D-20-01555R2 

Indigenous Australian women's experiences of participation in cervical screening 

Dear Dr. Butler:

I'm pleased to inform you that your manuscript has been deemed suitable for publication in PLOS ONE. Congratulations! Your manuscript is now with our production department. 

Kind regards, 

on behalf of

Dr. Nelly Oelke 

Academic Editor

PLOS ONE